# Thrombotic Microangiopathy among Hospitalized Patients with Systemic Lupus Erythematosus in the United States

**DOI:** 10.3390/diseases9010003

**Published:** 2020-12-24

**Authors:** Aleksandra I. Pivovarova, Charat Thongprayoon, Panupong Hansrivijit, Wisit Kaewput, Fawad Qureshi, Boonphiphop Boonpheng, Tarun Bathini, Michael A Mao, Saraschandra Vallabhajosyula, Wisit Cheungpasitporn

**Affiliations:** 1Division of Nephrology, Department of Internal Medicine, University of Mississippi Medical Center, Jackson, MS 39216, USA; apivovarova@umc.edu; 2Division of Nephrology and Hypertension, Department of Medicine, Mayo Clinic, Rochester, MN 55905, USA; Qureshi.Fawad@mayo.edu; 3Department of Internal Medicine, University of Pittsburgh Medical Center Pinnacle, Harrisburg, PA 17101, USA; hansrivijitp@upmc.edu; 4Department of Military and Community Medicine, Phramongkutklao College of Medicine, Bangkok 10400, Thailand; 5Department of Medicine, David Geffen School of Medicine, University of California, Los Angeles, CA 90095, USA; boonpipop.b@gmail.com; 6Department of Internal Medicine, University of Arizona, Tucson, AZ 85721, USA; tarunjacobb@gmail.com; 7Division of Nephrology and Hypertension, Department of Medicine, Mayo Clinic, Jacksonville, FL 32224, USA; mao.michael@mayo.edu; 8Section of Interventional Cardiology, Division of Cardiovascular Medicine, Department of Medicine, Emory University School of Medicine, Atlanta, GA 30322, USA; saraschandra.vallabhajosyula@emory.edu

**Keywords:** systemic lupus erythematosus, SLE, hospitalization, outcomes, risk factor, mortality

## Abstract

Background: This study aimed to evaluate thrombotic microangiopathy’s (TMA) incidence, risk factors, and impact on outcomes and resource use in hospitalized patients with systemic lupus erythematosus (SLE). Methods: We used the National Inpatient Sample to construct a cohort of hospitalized patients with SLE from 2003–2014. We compared clinical characteristics, in-hospital treatments, outcomes, and resource use between SLE patients with and without TMA. Results: Of 35,745 hospital admissions for SLE, TMA concurrently presented or developed in 188 (0.5%) admissions. Multivariable analysis showed that age ≥ 40 years and Hispanics were significantly associated with decreased risk of TMA, whereas Asian/Pacific Islanders and history of chronic kidney disease were significantly associated with increased risk of TMA. TMA patients required more kidney biopsy, plasmapheresis, mechanical ventilation, and renal replacement therapy. TMA was significantly associated with increased risk of in-hospital mortality and acute conditions including hemoptysis, glomerulonephritis, encephalitis/myelitis/encephalopathy, hemolytic anemia, pneumonia, urinary tract infection, sepsis, ischemic stroke, seizure, and acute kidney injury. The length of hospital stays and hospitalization cost was also significantly higher in SLE with TMA patients. Conclusion: TMA infrequently occurred in less than 1% of patients admitted for SLE, but it was significantly associated with higher morbidity, mortality, and resource use.

## 1. Introduction

Thrombotic microangiopathy (TMA) is a clinical syndrome described by the clinical presentation of thrombocytopenia, microangiopathic hemolytic anemia (MAHA), and evidence of organ injury [1]. Pathologically, TMA consists of microvascular thromboses in the vessel walls of arterioles and capillaries [2]. TMA can lead to significant morbidity and mortality. Renal involvement with acute kidney injury (AKI) that can progress to need for renal replacement therapy is one such common complication of TMA [1]. TMA can be hereditary or acquired. Acquired causes of TMA can be classified into ADAMTS13 deficiency-mediated, Shiga toxin-mediated, drug-mediated, and complement-mediated TMA [2]. Several systemic diseases have also been associated with TMA. Thrombocytopenic purpura (TTP) and hemolytic uremic syndrome (HUS) are the most commonly described hematological diseases associated with TMA [2]. Autoimmune diseases, such as systemic lupus erythematosus (SLE), scleroderma renal crisis, catastrophic antiphospholipid syndrome, or malignant hypertension can also cause TMA [2,3,4,5,6].

The concurrent presence of TMA in SLE patients has been reported in small series to range from approximately 1% to 24% [2,3,4,5,6,7,8,9]. Current evidence suggests that SLE-associated TMA may be driven by antiphospholipid syndrome, HELLP syndrome during pregnancy, TMA associated glomerulonephritis, or complement-mediated TMA [6,10,11]. Mortality in SLE patients who develop TMA has been reported to be as high as 31.9%, despite plasma exchange [12]. While SLE patients with TMA have a high mortality risk, data from large contemporary studies on hospitalized SLE patients with TMA were limited.

Thus, we conducted this study to evaluate TMA’s incidence, risk factors, and impact on outcomes and resource use in hospitalized SLE patients in the United States

## 2. Materials and Methods

### 2.1. Data Source

This cohort study utilized the National Inpatient Sample (NIS) database, the largest all-payer inpatient database in the United States. The Healthcare Cost and Utilization Project (HCUP) under the sponsorship of the Agency for Healthcare Research and Quality (AHRQ) manages and maintains the NIS database. The NIS database contains hospitalization data from a 20% stratified sample of hospitals in the United States. The NIS database does not examine individual patients, but rather single inpatient admissions. The data includes patient demographics, primary diagnosis, up to 24 secondary diagnoses, and procedural codes. Institutional review board approval was exempted because the NIS is a publicly available de-identified database.

### 2.2. Study Population

Using the NIS data from 2003 to 2014, we constructed a retrospective cohort of hospitalized patients with a primary discharge diagnosis of SLE based on International Classification of Diseases, Ninth Edition (ICD-9) diagnosis code of 710.0. We categorized SLE patients based on the presence of TMA during hospitalization. We identified TMA using ICD-9 diagnosis code of 446.6.

### 2.3. Data Collection

We identified clinical characteristics, treatments, and outcomes using ICD-9 codes, shown in Appendix A. Clinical characteristics consisted of age, sex, race, hypertension, dyslipidemia, chronic kidney disease, and cirrhosis. Procedures and treatments consisted of kidney biopsy, plasmapheresis, invasive mechanical ventilation, and renal replacement therapy. Complications and outcomes consisted of hemoptysis, pleural effusion/pleuritis, glomerulonephritis, encephalitis/myelitis/encephalopathy, hemolytic anemia, thrombocytopenia, pneumonia, urinary tract infection, sepsis, ischemic stroke, seizure, acute kidney injury, and in-hospital mortality. Resource use consisted of length of hospital stay and hospitalization cost.

### 2.4. Statistical Analysis

The difference in clinical characteristics, treatments, outcomes, and resource use between SLE patients with and without TMA was tested using student’s *t*-test for continuous variables and Chi-squared test for categorical variables. Multivariable logistic regression with backward stepwise selection was performed to evaluate independent risk factors for TMA. The associations of TMA with treatments and outcomes were evaluated using logistic regression analysis. The associations of TMA with resource use were evaluated using linear regression analysis. Analyses were adjusted for baseline clinical characteristics. Analysis results were statistically significant when two-tailed *p*-value < 0.05. SPSS statistical software (version 22.0, IBM Corporation, Armonk, NY, USA) was used for all analyses.

## 3. Result

### 3.1. TMA Incidence and Risk Factors in Hospitalized SLE Patients

Of 35,745 hospital admissions for SLE, 188 (0.5%) had a concurrent diagnosis of TMA during their hospitalization. Table 1 compares clinical characteristics, in-hospital treatments, outcomes, and resource use between SLE patients with and without TMA. SLE patients with TMA had significant higher in-hospital mortality than those without TMA (Figure 1). Multivariable analysis showed that age ≥ 40 years and Hispanics were significantly associated with decreased risk of TMA, whereas Asian/Pacific Islanders and history of chronic kidney disease were significantly associated with increased risk of TMA (Table 2).

### 3.2. Association of TMA in SLE patients with In-Hospital Treatments, Outcomes, and Resource Use

Kidney biopsy was performed in 21.3% of TMA compared to 12.4% of non-TMA SLE patients. Plasmapheresis was performed in 43.6% in TMA compared to 1.0% of non-TMA SLE patients. Renal replacement therapy was performed in 36.2% in TMA compared to 11.1% of non-TMA SLE patients. After adjusting for baseline characteristics, TMA was significantly associated with higher requirements of kidney biopsy (OR 1.64; *p* = 0.007), plasmapheresis (OR 67.44; *p* < 0.001), invasive mechanical ventilation (OR 7.27; *p* < 0.001), and renal replacement therapy (OR 5.01; *p* < 0.001) (Table 3).

In adjusted analyses, TMA in SLE patients was significantly associated with increased risk of several complications, including hemoptysis (OR 3.27; *p* < 0.001), glomerulonephritis (OR 5.25; *p* < 0.001), encephalitis/myelitis/encephalopathy (OR 2.92; *p* < 0.001), hemolytic anemia (OR 2.13; *p* < 0.001), pneumonia (OR 2.49; *p* < 0.001), urinary tract infection (OR 1.53; *p* = 0.04), sepsis (OR 2.99; *p* < 0.001), ischemic stroke (OR 6.30; *p* < 0.001), seizure (OR 2.16; *p* < 0.001), and acute kidney injury (OR 3.92; *p* < 0.001). TMA was significantly associated with a higher risk of in-hospital mortality compared to non-TMA SLE patients (OR 5.54; *p* < 0.001).

TMA in SLE patients was significantly associated with both a longer mean length of hospital stay with a difference of 12.4 days (*p* < 0.001) and higher mean hospitalization cost with a difference of $148,862 (*p* < 0.001).

## 4. Discussion

Multiple studies looked at the association of TMA with SLE, with reported prevalence of TMA among patient with SLE varying between 1% and 24% [2,3,4,5,6,7,8,9]. However, from the previous studies, the largest study included 285 patients [2,3,4,5,6,7,8,9]. It has been hypothesized that the difference in numbers was due to some diagnoses being made based on histopathology only, not taking in consideration clinical evaluation [6]. In this study, we reported an incidence of concurrent TMA of 0.5% among hospitalized patients with SLE. To date, our study is the largest cohort to describe the incidence of TMA in SLE patients. More importantly, we also described (1) the risk factors (Asian/Pacific Islanders and history of chronic kidney disease) and (2) the protective factors (age ≥ 40 years, and Hispanic race) for TMA in SLE patients for the first time.

A large-population study by Gomez Puerta was the first study to observed lower mortality rates among Hispanic patients with SLE compared to black, white and Native American, while most previous studies reported higher mortality among Blacks and Hispanics [13]. This study describes a term “Hispanic paradox” among SLE patients, claiming that after adjusting for age and annual income, Hispanic population with SLE had a lower mortality even though this population has a higher burden of SLE [13]. Our study revealed that Hispanic race is indeed associated with decreased risk of developing TMA in SLE patients. TMA, in its worst form, can lead to thrombi formation in glomeruli, arterioles and capillaries, causing decline in renal function [7]. Likewise, SLE exhibits renal manifestation in 50% of patients [14]. Studies by Chen at al. [9] and Letchumanan et al. [15] also revealed that patients with baseline renal impairment had lower survival rates than those with normal baseline renal function. Our study identified presence of CKD as a significant factor associated with TMA in SLE patients. Among other factors such as hypertension, dyslipidemia, and cirrhosis, only CKD showed to be statistically significant risk factor. Baseline CKD is a major risk factor for development of TMA in SLE. Besides CKD, Asian/Pacific islander race was also associated with higher risk of developing TMA in SLE. There are not enough studies at this time to elaborate on this association, however it was previously reported that mortality from SLE has been three to six times higher in Asians than Whites [16,17].

Previously, Barrera-Vargas et al. [18] attempted to identify possible risk factors contributing to TMA in SLE patients, and further define TMA’s impact on clinical outcomes. This study showed lymphopenia and anti-Ro positivity as independent risk factors for developing renal TMA. It also showed no significant difference in outcomes for patients with similar eGFR with or without TMA, and no significant effect on renal survival in TMA patients [18]. A study out of Northern Taiwan including 2461 SLE patients highlighted infection as a potential major trigger for TMA occurrence [9]. Our study findings identified other potential factors significantly associated with increased risk of developing TMA among hospitalized SLE patients. Increased risk was associated with Asian/Pacific islander race and history of CKD, while decreased risk with age ≥ 40 years and Hispanic race.

Prior studies have also investigated the impact of TMA on morbidity and mortality for SLE patients. Chen et al. previously reported that age > 40 years, acute kidney injury, presence of infection, TMA features and low complement C3 < 60 mg/dL were risk factors for increased mortality among 25 SLE patients with TMA [9]. In addition, Peigne et al. demonstrated that cardiac involvement in SLE-associated TMA patients was associated with increased mortality while plasma exchange was a significant protective factor [19]. The clinical outcomes of patients with SLE-associated TMA may be dependent on ADAMTS13 activity. Some studies have identified that a subgroup of SLE-associated TMA patients with acquired ADAMTS13 deficiency had a relatively benign outcome, even though these patients presented with extreme thrombocytopenia and CNS symptoms [20,21]. In line with our study, SLE-associated TMA was linked to complications in almost every organ with increased risk of hemoptysis, pneumonia, neurologic manifestations such as encephalitis/myelitis/encephalopathy, ischemic stroke, seizure, hemolytic anemia, UTI, sepsis, glomerulonephritis and AKI. We also identified that TMA was associated with higher requirements for kidney biopsy, plasmapheresis, invasive mechanical ventilation and renal replacement therapy. With the identified increased morbidity and mortality of SLE-associated TMA, it was not surprising that TMA was also associated with significantly increased length of hospital stay and cost. The length of hospital stays and cost in SLE-associated TMA patients was increased by 12.4 days and $148,862 in comparison to their non-TMA counterparts, respectively.

Our study has several limitations. First, although the NIS database is a large hospitalization database in the U.S., due to limitations of the database, it is impossible to determine the onset of the SLE diagnosis prior to hospitalization and the disease activity. Second, the diagnosis of TMA was identified by ICD-9 diagnosis code. Only 21.3% of SLE patients with TMA had a kidney biopsy in our study representing that majority of TMA was diagnosed by clinical diagnosis. In addition, laboratory data and kidney biopsy results were not reported by the NIS. It would be clinically relevant to know the activities of SLE by laboratory parameters, and if renal involvement in these patients was due to concurrent lupus nephritis or isolated TMA. Third, the incidence of TMA in SLE patients may be underestimated, as less severe cases may have been managed in the outpatient setting. It would thus follow that the associated factors and morbidity identified in this study is specific to SLE patients requiring hospitalization. Lastly, there was no data on drug treatment in the NIS database. Consequently, we could not assess the treatment effects of medications, such as eculizumab, on outcomes in SLE patients with TMA [22].

## 5. Conclusions

In conclusion, TMA infrequently occurred in less than 1% of patients admitted for SLE, but it was associated with a significantly higher morbidity, mortality, and resource use. History of chronic kidney and Asian/Pacific Islanders were significantly associated with increased risk of TMA while age ≥ 40 years and Hispanics were significantly associated with decreased risk of TMA.

## Figures and Tables

**Figure 1 diseases-09-00003-f001:**
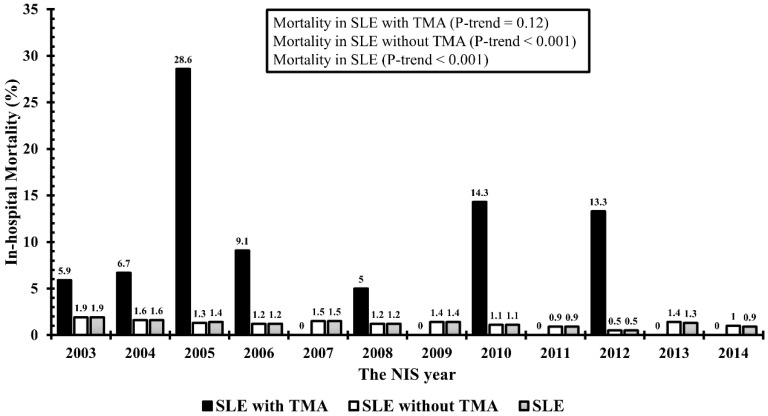
In-hospital mortality among SLE patients with TMA vs without TMA.

**Table 1 diseases-09-00003-t001:** Clinical characteristics, in-hospital treatments, complications, outcomes, and resource utilization in SLE patients with and without TMA.

	Total	SLE with TMA	SLE without TMA	*p*-Value
Clinical characteristics				
N (%)	35,745	188 (0.5)	35,557 (99.5)	
Age (years), mean (SD)	35.8 (15.9)	31.7 (14.8)	35.8 (15.9)	<0.001
<20	5595 (15.7)	37 (19.7)	5558 (15.6)	0.02
20–29	8942 (25.0)	60 (31.9)	8882 (25.0)	
30–39	7646 (21.4)	41 (21.8)	7605 (21.4)	
40–49	6458 (18.1)	25 (13.3)	6433 (18.1)	
≥50	7087 (19.8)	25 (13.3)	7062 (19.9)	
Male	4732 (13.3)	29 (15.4)	4703 (13.2)	0.38
Race				
Caucasian	8577 (24.0)	47 (25.0)	8530 (24.0)	<0.001
African American	13,156 (36.8)	66 (35.1)	13,090 (36.8)	
Hispanic	6103 (17.1)	17 (9.0)	6086 (17.1)	
Asian or Pacific Islander	1570 (4.4)	22 (11.7)	1548 (4.4)	
Other	6339 (17.7)	36 (19.1)	6303 (17.7)	
Hypertension	17,174 (48.0)	103 (54.8)	17,071 (48.0)	0.06
Dyslipidemia	2965 (8.3)	13 (6.9)	2952 (8.3)	0.49
Chronic kidney disease	5989 (16.8)	53 (28.2)	5936 (16.7)	<0.001
Cirrhosis	1275 (3.6)	11 (5.9)	1264 (3.6)	0.09
Management				
Kidney biopsy	414 (1.2)	40 (21.3)	4401 (12.4)	<0.001
plasmapheresis	444 (1.2)	82 (43.6)	362 (1.0)	<0.001
Invasive mechanical ventilation	897 (2.5)	32 (17.0)	865 (2.4)	<0.001
Renal replacement therapy	4002 (11.2)	68 (36.2)	3934 (11.1)	<0.001
Complication and outcomes				
Hemoptysis	688 (1.9)	12 (6.4)	676 (1.9)	<0.001
Pleural effusion/pleuritis	3354 (9.4)	17 (9.0)	3337 (9.4)	0.87
Glomerulonephritis	18,201 (50.9)	159 (84.6)	18,042 (50.7)	<0.001
Encephalitis/myelitis/encephalopathy	2351 (6.6)	33 (17.6)	2318 (6.5)	<0.001
Antiphospholipid syndrome	1882 (5.3)	22 (11.7)	1860 (5.2)	<0.001
Hemolytic anemia	6341 (17.7)	65 (34.6)	6276 (17.7)	<0.001
Thrombocytopenia	3474 (9.7)	21 (11.2)	3453 (9.7)	0.50
Pneumonia	2775 (7.8)	32 (17.0)	2743 (7.7)	<0.001
Urinary tract infection	3689 (10.3)	27 (14.4)	3662 (10.3)	0.07
Sepsis	1143 (3.2)	18 (9.6)	1125 (3.2)	<0.001
Ischemic stroke	727 (2.0)	21 (11.2)	706 (2.0)	<0.001
Seizure	2990 (8.4)	33 (17.6)	2957 (8.3)	<0.001
Acute kidney injury	5404 (15.1)	81 (43.1)	5323 (15.0)	<0.001
In-hospital mortality	455 (1.3)	12 (6.4)	443 (1.2)	<0.001
Resource utilization				
Length of hospital stay (days), mean (SD)	6.6 (8.6)	19.2 (17.2)	6.5 (8.5)	<0.001
Hospitalization cost ($), mean (SD)	43,825.3 (82769.4)	196,213.5 (210558.5)	43,021.0 (80,814.5)	<0.001

**Table 2 diseases-09-00003-t002:** Factors associated with TMA in SLE patients.

Variables	Univariable Analysis		Multivariable Analysis	
	Crude Odds Ratio (95%CI)	*p*-Value	Adjusted Odds Ratio (95%CI)	*p*-Value
Age (years)				
<20	1 (reference)		1 (reference)	
20–29	1.02 (0.67–1.53)	0.94	1.00 (0.66–1.51)	0.99
30–39	0.81 (0.52–1.27)	0.35	0.80 (0.51–1.26)	0.34
40–49	0.58 (0.35–0.97)	0.04	0.58 (0.35–0.97)	0.04
≥50	0.53 (0.32–0.88)	0.02	0.49 (0.29–0.82)	0.007
Male	1.20 (0.80–1.78)	0.38		
Race				
Caucasian	1 (reference)		1 (reference)	
African American	0.92 (0.63–1.33)	0.64	0.75 (0.51–1.10)	0.13
Hispanic	0.51 (0.29–0.88)	0.02	0.39 (0.22–0.69)	0.001
Asian or Pacific Islander	2.58 (1.55–4.29)	<0.001	1.99 (1.18–3.35)	0.009
Other	1.04 (0.67–1.60)	0.87	0.89 (0.57–1.38)	0.61
Hypertension	1.31 (0.98–1.75)	0.06		
Dyslipidemia	0.82 (0.47–1.44)	0.49		
Chronic kidney disease	1.96 (1.42–2.70)	<0.001	2.04 (1.48–2.82)	<0.001
Cirrhosis	0.59 (0.32–1.09)	0.09		

**Table 3 diseases-09-00003-t003:** The association between TMA in SLE patients and in-hospital treatment, complications, outcomes, resource utilization.

	Univariable Analysis		Multivariable Analysis	
	Crude Odds Ratio (95% CI)	*p*-Value	Adjusted Odds Ratio* (95% CI)	*p*-Value
Treatments				
Kidney biopsy	1.91 (1.35–2.72)	<0.001	1.64 (1.14–2.34)	0.007
Therapeutic plasmapheresis	75.21 (55.37–102.16)	<0.001	67.44 (49.12–92.60)	<0.001
Invasive mechanical ventilation	8.23 (5.59–12.11)	<0.001	7.27 (4.90–10.79)	<0.001
Renal replacement therapy	4.56 (3.38–6.15)	<0.001	5.01 (3.43–7.31)	<0.001
Complications and outcomes				
Hemoptysis	3.52 (1.95–6.35)	<0.001	3.27 (1.80–5.91)	<0.001
Pleural effusion/pleuritis	0.96 (0.58–1.58)	0.87	0.92 (0.56–1.52)	0.75
Glomerulonephritis	5.32 (3.58–7.91)	<0.001	5.25 (3.39–8.14)	<0.001
Encephalitis/myelitis/encephalopathy	3.05 (2.09–4.46)	<0.001	2.92 (1.99–4.26)	<0.001
Antiphospholipid syndrome	2.40 (1.54–3.76)	<0.001	2.31 (1.47–3.63)	<0.001
Hemolytic anemia	2.47 (1.82–3.33)	<0.001	2.13 (1.55–2.92)	<0.001
Thrombocytopenia	1.17 (0.74–1.84)	0.50	1.00 (0.63–1.58)	1.00
Pneumonia	2.45 (1.67–3.60)	<0.001	2.49 (1.69–3.66)	<0.001
Urinary tract infection	1.46 (0.97–2.20)	0.07	1.53 (1.01–2.31)	0.04
Sepsis	3.24 (1.99–5.29)	<0.001	2.99 (1.82–4.90)	<0.001
Ischemic stroke	6.21 (3.92–9.83)	<0.001	6.30 (3.96–10.02)	<0.001
Seizure	2.35 (1.61–3.42)	<0.001	2.16 (1.48–3.17)	<0.001
Acute kidney injury	4.30 (3.22–5.75)	<0.001	3.92 (2.90–5.30)	<0.001
In-hospital mortality	5.40 (2.99–9.77)	<0.001	5.54 (3.03–10.15)	<0.001
	Coefficient (95% CI)	*p*-value	Adjusted coefficient * (95% CI)	*p*-value
Resource utilization				
Length of hospital stay (days)	12.7 (11.5–13.9)	<0.001	12.4 (11.2–13.6)	<0.001
Hospitalization cost ($)	153,192.5 (141,308.8–165,076.2)	<0.001	148,862.0 (137,111.8–160,612.0)	<0.001

* Adjusted for age, sex, race, hypertension, dyslipidemia, chronic kidney disease, and cirrhosis.

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
