# Peer review of "Thrombotic Microangiopathy among Hospitalized Patients with Systemic Lupus Erythematosus in the United States"

_diseases, 2020, doi:10.3390/diseases9010003_

Round 1

Reviewer 1 Report

Pivovarova et al submit  a paper entitled "Thrombotic Microangiopathy among Hospitalized Patients with Systemic Lupus Erythematosus in the United States". In htis retrospective cohort study, they looked at thrombotic microangiopathy’s (TMA) incidence, risk factors, and impact on outcomes and resource use in hospitalized patients with systemic lupus erythematosus (SLE).

Of 35,745 hospital admissions for SLE, TMA concurrently presented or developed in 188 (0.5%) admissions. in these patients, SLE was significantly associated with higher morbidity, mortality, and resource use.

The National Inpatient Sample (NIS) database was used. Institutional review board approval was exempted because the NIS is a publicly available de-identified database.

This study is mainly conformatory as a link between SLE and TMI has been amply demonstrated.  It is well written and well presented, but lacks summarizing figures.

Minor

line 154 : rewrite "while decreased risk with age =60 years and
Hispanic race"

Author Response

Reviewer 1

Pivovarova et al submit  a paper entitled "Thrombotic Microangiopathy among Hospitalized Patients with Systemic Lupus Erythematosus in the United States". In this retrospective cohort study, they looked at thrombotic microangiopathy’s (TMA) incidence, risk factors, and impact on outcomes and resource use in hospitalized patients with systemic lupus erythematosus (SLE).

Response: We thank you for reviewing our manuscript and for your critical evaluation. 

Comment #1

Of 35,745 hospital admissions for SLE, TMA concurrently presented or developed in 188 (0.5%) admissions. in these patients, SLE was significantly associated with higher morbidity, mortality, and resource use. The National Inpatient Sample (NIS) database was used. Institutional review board approval was exempted because the NIS is a publicly available de-identified database. This study is mainly confirmatory as a link between SLE and TMA has been amply demonstrated.  It is well written and well presented, but lacks summarizing figures.

Response:  We appreciate the reviewer’s important point. We thus additionally included Figure 1 to summarize in-hospital mortality rate among SLE patients with TMA vs without TMA in the result of manuscript as suggested.

Comment #2

                line 154 : rewrite "while decreased risk with age =60 years and Hispanic race"

Response:  We appreciate the reviewer’s thorough review and apologize for an error. We have made this correction as suggested. 

We greatly appreciated the reviewers’ and editors’ time and comments to improve our manuscript. The manuscript has been improved considerably by the suggested revisions.

Reviewer 2 Report

My comments and questions: 

Please review the table 1. again, the p values are not in the right place.  

Why was the study completed in 2014? Do you know something in the last 5 years? 

The number of kidney biopsies seems too low in both groups, what is the reason?   

Do you know something about the presence of anti-phospholipid antibodies and antiphospholipid syndrome?

It would have been lucky if we had known more about the activity of the lupus and some immunological parameters. 

Author Response

Reviewer 2

Comment #1

My comments and questions:

Please review the table 1. again, the p values are not in the right place. 

Response: We thank you for reviewing our manuscript and for your critical evaluation.  We appreciate the reviewer’s important point.  We have reviewed and corrected our p-values as the reviewer’s suggestion.

Comment #2

Why was the study completed in 2014? Do you know something in the last 5 years?

Response:  We appreciate the reviewer’s important point.  We apologize we have database available until 2014. In 2015, NIS started the ICD-10 implementation and potentially made discrepancies and heterogeneity of the results.

Comment #3

                The number of kidney biopsies seems too low in both groups, what is the reason?  

Response:  The reviewer made an important and very good point. In our study, TMA was identified by ICD code in the database. Low number of kidney biopsies in the study potentially represents that majority of TMA cases were diagnosed by clinical diagnosis of TMA rather than a kidney biopsy. We appreciate the reviewer’s important comment that thus additionally included this important point in the limitation of our study as suggested.

“Our study has several limitations. First, although the NIS database is a large hospitalization database in the U.S., due to limitations of the database, it is impossible to determine the onset of the SLE diagnosis prior to hospitalization and the disease activity. Second, the diagnosis of TMA was identified by ICD-9 diagnosis code. Only 21.3% of SLE patients with TMA had a kidney biopsy in our study representing that majority of TMA was diagnosed by clinical diagnosis. In addition, laboratory data and kidney biopsy results were not reported by the NIS. It would be clinically relevant to know the activities of SLE by laboratory parameters, and if renal involvement in these patients was due to concurrent lupus nephritis or isolated TMA. Third, the incidence of TMA in SLE patients may be underestimated, as less severe cases may have been managed in the outpatient setting. It would thus follow that the associated factors and morbidity identified in this study is specific to SLE patients requiring hospitalization. Lastly, there was no data on drug treatment in the NIS database. Consequently, we could not assess the treatment effects of medications, such as eculizumab, on outcomes in SLE patients with TMA [22].” 

Comment #4

Do you know something about the presence of anti-phospholipid antibodies and antiphospholipid syndrome?

Response:  We agree with this important point. We have thus additionally included data on antiphospholipid and incorporated in our data analysis and manuscript as reviewer’s suggestion.

Comment #5

It would have been lucky if we had known more about the activity of the lupus and some immunological parameters.

Response:  The reviewer’s point is important. We agree with this point; however, data on laboratory data and kidney biopsy findings were limited in the NIS database. We have added this important point in the limitation as suggested.  

Second, the diagnosis of TMA was identified by ICD-9 diagnosis code. Only 21.3% of SLE patients with TMA had a kidney biopsy in our study representing that majority of TMA was diagnosed by clinical diagnosis. In addition, laboratory data and kidney biopsy results were not reported by the NIS. It would be clinically relevant to know the activities of SLE by laboratory parameters, and if renal involvement in these patients was due to concurrent lupus nephritis or isolated TMA.

We greatly appreciated the reviewers’ and editors’ time and comments to improve our manuscript. The manuscript has been improved considerably by the suggested revisions.

Reviewer 3 Report

Considering with TMA, organ injuries may be important. Did they analyze in(hospitalized)-patients due to SLE? As a critical point, how could authors decide cases without TMA in cases without renal biopsies? If renal biopsies were performed for every patient, please state them in the “Materials and Methods”, clearly. At that case, it would be better to change “analysis of lupus nephritis” from “analysis of SLE” to specify the present study clearly. If authors involved other organ failures without renal injuries, in which type of tissue biopsy did they diagnose TMA?

 Therefore, authors should categorize their cases according to complication of tissue injuries, for example, with/without lupus nephritis, neuro lupus, and/or the other organ failures. Concerning with nephritis, it would be better to divided their cases into acute renal failure and chronic renal failure.

  1. Why did authors analyze between 60 yo. ≧ and >60 yo. to evaluate risk factors to develop TMA? Authors should describe some adequate reasons to decide.
  2. Again, authors focused line of 60 yo.. However, in table 1, authors categorized their population into different ages. Why?
  3. Different diagnosis of SLE should be stated in the “Materials and Methods” in the present cases.
  4. How about disease activity such as C3, C4 and/or CH50, at least, and incidence of TMA in lupus nephritis which authors showed significant organ failure in SLE population. I guess that authors can evaluate tissue activity under renal pathology.
  5. Discussion is also weak. Although they appealed ages and species for incidence of TMA, their discussion did not fully support their results. It would be recommended to reconsider their discussion according to their results.

Author Response

Reviewer 3

Comment #1

Considering with TMA, organ injuries may be important. Did they analyze in(hospitalized)-patients due to SLE? As a critical point, how could authors decide cases without TMA in cases without renal biopsies? If renal biopsies were performed for every patient, please state them in the “Materials and Methods”, clearly. At that case, it would be better to change “analysis of lupus nephritis” from “analysis of SLE” to specify the present study clearly. If authors involved other organ failures without renal injuries, in which type of tissue biopsy did they diagnose TMA? Therefore, authors should categorize their cases according to complication of tissue injuries, for example, with/without lupus nephritis, neuro lupus, and/or the other organ failures. Concerning with nephritis, it would be better to divided their cases into acute renal failure and chronic renal failure.

Response: We thank you for reviewing our manuscript and for your critical evaluation.  The reviewer raised very important point. We constructed a retrospective cohort of hospitalized patients with a primary discharge diagnosis of SLE based on International Classification of Diseases, Ninth Edition (ICD-9) diagnosis code of 710.0. We categorized SLE patients based on the presence of TMA during hospitalization. We identified TMA using ICD-9 diagnosis code of 446.6.

The reviewer made an important and very good point. In our study, TMA was identified by ICD code in the database. Low number of kidney biopsies in the study potentially represents that majority of TMA cases were diagnosed by clinical diagnosis of TMA rather than a kidney biopsy. We appreciate the reviewer’s important comment that thus additionally included this important point in the limitation of our study as suggested.

“Our study has several limitations. First, although the NIS database is a large hospitalization database in the U.S., due to limitations of the database, it is impossible to determine the onset of the SLE diagnosis prior to hospitalization and the disease activity. Second, the diagnosis of TMA was identified by ICD-9 diagnosis code. Only 21.3% of SLE patients with TMA had a kidney biopsy in our study representing that majority of TMA was diagnosed by clinical diagnosis. In addition, laboratory data and kidney biopsy results were not reported by the NIS. It would be clinically relevant to know the activities of SLE by laboratory parameters, and if renal involvement in these patients was due to concurrent lupus nephritis or isolated TMA. Third, the incidence of TMA in SLE patients may be underestimated, as less severe cases may have been managed in the outpatient setting. It would thus follow that the associated factors and morbidity identified in this study is specific to SLE patients requiring hospitalization. Lastly, there was no data on drug treatment in the NIS database. Consequently, we could not assess the treatment effects of medications, such as eculizumab, on outcomes in SLE patients with TMA [22].” 

Comment #2

Why did authors analyze between 60 yo. ≧ and >60 yo. to evaluate risk factors to develop TMA? Authors should describe some adequate reasons to decide. Again, authors focused line of 60 yo.. However, in table 1, authors categorized their population into different ages. Why?

Response:  We appreciate the reviewer’s important point. We apologize for our error in typing of age group in the Tables. We have made correction of age groups/ranges in Tables as reviewer’s suggestion. Our

Comment #3

                Different diagnosis of SLE should be stated in the “Materials and Methods” in the present cases.

Response:  We appreciate the reviewer’s thorough review. We agree with the reviewer regarding this importance. However, ICD-9 diagnosis codes for different SLE diagnosis are not specifically provided for all organ involvements, and it is the limitation of the database. Thus, we explored outcomes of all organ involvements in Table 1 and Table 3.

Comment #4

How about disease activity such as C3, C4 and/or CH50, at least, and incidence of TMA in lupus nephritis which authors showed significant organ failure in SLE population. I guess that authors can evaluate tissue activity under renal pathology.

Response:  The reviewer raises important point. . We agree with this point; however, data on laboratory data and kidney biopsy findings were limited in the NIS database. We have added this important point in the limitation as suggested.  

Second, the diagnosis of TMA was identified by ICD-9 diagnosis code. Only 21.3% of SLE patients with TMA had a kidney biopsy in our study representing that majority of TMA was diagnosed by clinical diagnosis. In addition, laboratory data and kidney biopsy results were not reported by the NIS. It would be clinically relevant to know the activities of SLE by laboratory parameters, and if renal involvement in these patients was due to concurrent lupus nephritis or isolated TMA.

Comment #5

                Discussion is also weak. Although they appealed ages and species for incidence of TMA, their discussion did not fully support their results. It would be recommended to reconsider their discussion according to their results.

Response:  We apologize and we agree with the reviewer. We have now revised our discussion comprehensively to reflect and discuss on the findings of our study as suggested.

“A large-population study by Gomez Puerta was the first study to observed lower mortality rates among Hispanic patients with SLE compared to black, white and Native American, while most previous studies reported higher mortality among Blacks and Hispanics [13]. This study describes a term “Hispanic paradox” among SLE patients, claiming that after adjusting for age and annual income, Hispanic population with SLE had a lower mortality even though this population has a higher burden of SLE [13].

Our study revealed that Hispanic race is indeed associated with decreased risk of developing TMA in SLE patients. TMA , in its worst form, can lead to thrombi formation in glomeruli, arterioles and capillaries, causing decline in renal function [7]. Likewise, SLE exhibits renal manifestation in 50% of patients [14]. Studies by Chen at al. [9] and Letchumanan et al. [15] also revealed that patients with baseline renal impairment had lower survival rates than those with normal baseline renal function. Our study identified presence of CKD as a significant factor associated with TMA in SLE patients. Among other factors such as hypertension, dyslipidemia, and cirrhosis, only CKD showed to be statistically significant risk factor. Baseline CKD is a major risk factor for development of TMA in SLE. Besides CKD, Asian/Pacific islander race was also associated with higher risk of developing TMA in SLE. There are not enough studies at this time to elaborate on this association, however it was previously reported that mortality from SLE has been three to six times higher in Asians than Whites [16,17].”

We greatly appreciated the reviewers’ and editors’ time and comments to improve our manuscript. The manuscript has been improved considerably by the suggested revisions.

Round 2

Reviewer 2 Report

None

Reviewer 3 Report

I think authors responded all of my comments.